# Spanish Cultural Adaptation and Inter-Rater Reliability of the Revised Knox Preschool Play Scale

**DOI:** 10.3390/children10060965

**Published:** 2023-05-29

**Authors:** Sergio Serrada-Tejeda, Marta Pérez-de-Heredia-Torres, Susan H. Knox, Patricia Sánchez-Herrera-Baeza, Rosa M. Martínez-Piédrola, Paula Obeso-Benítez, Sergio Santos-del-Riego

**Affiliations:** 1Department of Physical Therapy, Occupational Therapy, Rehabilitation and Physical Medicine, Rey Juan Carlos University, Avenida de Atenas s/n., 28922 Alcorcón, Spain; 2Therapy in Action, Tarzana, CA 91356, USA; 3Department of Physical Therapy, Medicine and Biomedical Sciences, Universidade da Coruña, 15001 A Coruña, Spain

**Keywords:** children, cultural adaptation, childhood play, assessment, revised Knox preschool play scale

## Abstract

Background: The Revised Knox Preschool Play Scale (RKPPS) is a comprehensive assessment test that observes the level of play development; however, there is no culturally adapted version available with stable psychometric values that would allow its widespread use and provide objective information during clinical evaluations. Methods: Cross-cultural adaptation included direct and retrospective translations, along with cognitive interviews with pediatric occupational therapists to analyze the comprehensibility of the translation. In addition, a final phase of linguistic revision was carried out to determine the grammatical and semantic fit of the adapted version. Finally, inter-rater reliability was analyzed in a sample of typically developing children aged four to six years old. Results: The processes of translation and back-translation, cognitive interview, and linguistic review determined an adequate grammatical and semantic equivalence to the Spanish cultural context. Almost perfect agreement, with values between 0.82 and 0.94, was obtained for items and play dimensions, indicating that the precision of the measurements between both evaluators was excellent. Conclusions: The cross-culturally adapted version of the RKPPS meets the necessary adjustments for the sociocultural context and can be used in the clinical practice of occupational therapy.

## 1. Introduction

In childhood, play is an essential occupation and represents one of the central domains for occupational therapy (OT) practitioners [1,2]. In the early stages of development, through play the child learns and develops early skills. In addition, through exploration and imitation during play, interactions with the physical and social environment enable cognitive, motor, and social learning [3,4].

Historically, professionals assess play from a functional point of view, recognizing play as a sign to observe and quantify changes in children’s development skills [5,6,7]. However, although assessing play is important, it is not an easy construct to measure, given that its multiple dimensions and conceptions make it difficult to identify observable and quantifiable aspects that facilitate its assessment.

As formal assessments did not emerge until the 1970s, the occupational therapy community of these decades focused on clinical observation of play behavior [8], given the difficulties in conceptualization, the need for a natural setting for assessment, and the limited availability of assessment tools. These circumstances have led to the current prevalence of practitioners employing non-standardized tests and informal listings of play skills or observation of play [9,10,11].

Despite the value of play as an occupation, in occupational therapy, play has been considered primarily as a means of treatment and has been recognized as an observable manifestation of child development [9]. Therefore, therapists need assessment tests to evaluate developmental competence and detect changes in the child. To achieve these goals, occupational therapists conducting play observations need a “systematic way of observing play behaviors in order to determine child strengths, skills and play deficits” [12].

One of the assessments that provide a picture of the developmental level of children’s play age in OT is the Revised Knox Preschool Play Scale (RKPPS) [13]. This useful tool allows us to reflect the level of all areas of child development through observation in a natural environment, and its application is useful to evaluate those children who, due to their neurodevelopmental condition, cannot be evaluated with standardized tests. The RKPPS provides a description of typical play development that allows the therapist to code each child’s play according to observed parameters and to average the age of identified play skills. The resulting age and play profile allow intervention planning.

The RKPPS was initially developed in an English-speaking country and currently, no cross-cultural adaptation processes have been developed to facilitate its use in another linguistic and cultural context. Cultural adaptation is not a meaningless process, but a thorough method that prevents errors and avoids wasting researcher resources. Moreover, cross-cultural comparison of adapted tests is essential to ensure comparability of results due to the multicultural contexts of today’s society. Furthermore, for RKPPS to provide effective information in clinical assessments, culturally adapted versions are required to maintain the meaning of the test and have appropriate metrics to allow its widespread use. The aim of this study is to develop a cross-cultural adaptation process of the RKPPS to Spanish, to maintain the linguistic and conceptual equivalence of the original version, and to analyze inter-rater agreement to improve their clinical usefulness due to subjectivity.

## 2. Materials and Methods

This study was conducted in Spain and the results provided are part of a broader project aimed at assessing play in autistic children. To culturally adapt the test, the principles established by the International Test Commission (ITC) [14] for the translation and adaptation of tests and assessment instruments were considered, as well as to establish the equivalence of scores between linguistic and/or cultural groups. In addition, as suggested by the methodological proposal of Ramada-Rodilla et al. [15], a translation and back-translation phase, a cognitive interview and a final linguistic review were included. The Ethics Committee of the Universidad Rey Juan Carlos approved the study.

### 2.1. Participants

Four teams were formed for the study phases consisting of 3 groups of participants and an individual reviewer (Figure 1). The first, made up of bilingual translators (*n* = 5), was organized into different groups: a team of translators (*n* = 3), who performed the direct translation, two bilingual translators, and a bilingual occupational therapist. A second back-translation team was made up of expert bilingual translators (*n* = 2) different from the direct translation team. For the cognitive interview phase, the second group was obtained by convenience sampling and consisted of occupational therapists (*n* = 8). The third group was a committee of experts (*n* = 3) responsible for analyzing and accepting the translated versions in the previous processes of direct and back-translation. Finally, a single person, an expert in linguistics with a background in English literature and education, supervised the Spanish version of the test (*n* = 1).

For this study, participants had to reside in Spain. The occupational therapists recruited for the cognitive interviews had to be graduates in OT, have experience in pediatrics and with children over 3 years of age, and be Spanish nationals. The panel of experts consisted of a rehabilitation physician and 2 occupational therapists, 1 of them with experience in pediatric intervention.

For the inter-rater reliability analysis phase, 2 experienced pediatric occupational therapists were included. At this stage, the culturally adapted version of the RKPPS was used to evaluate the play recordings in a sample of 30 typically developing children between the ages of 4 years and 6 years and 11 months.

### 2.2. Measures

Revised Knox Preschool Play Scale (RKPPS) [13]: The RKPPS is a standardized instrument for assessing play in children from birth to 6 years [16] and correlating it with an approximate developmental age. This test describes play skills in 6-month intervals during the first 3 years of development, and in 1-year increments from ages 4 to 6 years. Although the scale requires two 30-min observations of the child in different environments, 1 indoors and 1 outdoors, adequate reliability has also been assessed by observing two 15-min periods [17]. The scale consists of 4 dimensions and 12 categories assessing space management, material management, pretense/symbolism, and participation. To score each category, the evaluator assigns the most representative age level based on the child’s observed play behaviors. The final scores reflect the child’s overall play age after averaging the scores for each category. Adequate evidence of inter-rater reliability was obtained for 15-min observation periods [17].

### 2.3. Procedure

#### 2.3.1. Direct Translation

For this phase, the original version was sent to the first team of translators, who translated the test independently. After obtaining each version, the panel of experts analyzed and produced the first draft. Subsequently, the members of the forward translation team analyzed and scored its conceptual equivalence [18,19] by identifying the following options: (A) similar translation; (B) moderate equivalence; and (C) no conceptual equivalence. When scores of B or C were assigned, the translations were sent back to the direct translation team to draft an alternative translation, which would be reviewed again by the expert committee.

#### 2.3.2. Back Translation

A back-translation process [20,21] was carried out in which the second group, consisting of 2 bilingual translators, independently performed the back-translation into English. In this phase, the group of translators identified the conceptual equivalence of the translated versions, which would then be revised by the panel of experts in case the translations did not match adequately. This process allows the creation of the new draft translation that will be used in the next phase.

#### 2.3.3. Cognitive Interviewing

At this stage, semi-structured cognitive interviews were conducted by retrospective verbal probing to assess the cultural relevance, equivalence, and comprehensibility of the obtained version. This methodology allows us to understand the way in which the addressees understand and process the information of the evaluation materials and items. To this end, the methodology for cognitive interviews proposed by Willis [22] was followed, which suggests a sample of between 5 and 10 participants and the use of verbal recordings to facilitate the analysis of the interviewees’ comments [22,23]. During the interviews, the following questions were asked: “How do you interpret what the item states?” and “If necessary, would you make any changes to improve the meaning of the item?”. Individually, and in accordance with Willis [22], participants were asked to “think aloud”, using, for example, paraphrasing or exemplification, to detect difficulties with semantics and grammar. In addition, the recommendations of Román-Oyola and Reynolds [18], indicate that if comprehension difficulties are identified by 2 or more respondents, the expert committee should revisit the item for easier understanding.

#### 2.3.4. Linguist Expert Revision

For this phase, the linguistic expert reviewed the adequacy of the comprehension difficulties and their adaptation by the research team to ensure agreement with the context of the target population.

#### 2.3.5. Inter-Rater Reliability

To participate in this phase, the main caregivers of each child signed the informed consent form in order to film a video recording of 15 min of play in an outdoor space and 15 min of play in an indoor space. The recording was made at the participant’s home and in the school playground. In the indoor space where the observations were made, toys were arranged to facilitate manipulation, assembly, and construction, as well as toys with sound, action figures, dolls, or kitchenettes to facilitate the observation of functional play. For outdoor space, assessment structures such as slides, swings, construction materials, rockers, rope ladders, tricycles, and ride on cars were available. Video recordings necessary to evaluate play observation with the RKPPS were made by the principal investigator (PI) (first author). Subsequently, each of the recordings was evaluated and scored in a space provided at the Faculty of Health Sciences of the Universidad Rey Juan Carlos.

For sample size calculation, the estimates and contingency tables suggested by Bujang and Baharum (2017) [24] were taken into consideration based on the prespecified values of statistical power, type I error (alpha), and effect size. For the latter, the number of RKKPS scale score categories as well as the specific coefficients for each rater (κ_1_ and κ_2_) were preset before starting the study. The values for each rater were adjusted, finally specifying values of 0.50 for κ_1_ and 0.80 for κ_2_. Taking into account these considerations and assuming a statistical test power of 0.80 and a type I error α = 0.05 (95%), a minimum of between 24 and 31 participants was required for the present study. Considering that the RKPPS test describes play skills in 1-year increments for age ranges between 4 and 6 years, the sample of participants (*n* = 30), was divided into 3 different groups: the first (*n* = 10) from 4 to 4–11 years old and a mean age of 4.6 ± 0.3 years; the second (*n* = 10) from 5 to 5–11 years and a mean age of 5.5 ± 0.2 years, and the third (*n* = 10) from 6 to 6–11 years old and a mean age of 6.2 ± 0.4 years.

Inter-rater analysis was performed with the statistical program IBM SPSS Statistics for Windows, V.27.0 (Copyright 2013 IBM SPSS Corp). Cohen’s Kappa Index (κ) is a quantitative measure of 2-rater reliability that allows us to determine the level of inter-rater agreement on a set of items, correcting for the degree of agreement that may be due to chance and assigning a standardized index of inter-rater reliability (Hallgren, 2012) [25]. For the interpretation of the results of the κ index, the ranges suggested by Landis and Koch [26] were used: 0.00–0.20 (minimum agreement); 0.21–0.40 (correct agreement); 0.41–0.60 (moderate agreement); 0.61–0.80 (strong agreement) and 0.81–1.00 (almost perfect agreement). Since scores below 0.40 are poor, this study performed a review and a new inter-rater reliability process on all those items for which κ values below 0.50 were detected. This score was determined with the aim of guaranteeing greater rigor to the reliability process. Differences were considered statistically significant if *p* < 0.05.

## 3. Results

### 3.1. Direct and Back Translation

The results of direct translation indicate that 85% of the items were similar to the translation performed by the first group of bilingual translators (A) while 15% showed a moderate equivalence (B). After back-translation, adequate equivalence (A) was identified in 90% of the items. The remaining 10% showed moderate equivalence (B). The panel of experts agreed on a single translation when the use of expressions that hindered the comprehensibility of the translated text was identified. Table 1 describes the process carried out for some items.

### 3.2. Cognitive Interviews

The occupational therapists interviewed showed adequate comprehension, being able to rephrase or give an example of the translated items. However, 18 of the items showed some difficulty in comprehensibility due to the use of certain words or the syntactic organization of the item. To ensure a better fit, these items were reviewed and modified by the committee of experts. Table 2 shows some examples of the changes made to some of the items in each of the play dimensions.

### 3.3. Expert Linguist Review

The modifications suggested in the previous phases were reviewed and included several changes that ensured the necessary grammatical and linguistic concordance for the application context. In addition, since this is an evaluation test for children, to avoid sexism in some items the expert indicated the use of inclusive language through sentences with impersonal and reflexive passive forms, since in Spanish these syntactic constructions lack a grammatical subject, and in this way many subjects constructed in generic masculine are avoided. This type of adjustment was made, for example, to modify item “Child feeding doll”, which was initially translated as “El niño alimenta a una muñeca” and after a linguistic review it was suggested to make an adjustment to avoid sexist language, since the term “niño” in Spanish is masculine. The resulting version after the changes was “Se da de comer a una muñeca”.

### 3.4. Inter-Rater Reliability

The degree of inter-rater agreement for the RKPPS, calculated using the κ coefficient, obtained scores above 0.50 for the items evaluated, so that the precision of the measurements made was adequate and no changes to the items were necessary (Table 3). The set of items forming each dimension obtained scores associated with an almost perfect degree of correlation, with values between 0.82 and 0.94, indicating that the precision of the measurements between both evaluators was excellent.

Table 4 shows the κ results of the total scores of the four dimensions and the total score. All dimensions, as well as the RKKPS total score, obtained values above 0.80, indicating an excellent degree of agreement between the two evaluators.

## 4. Discussion

This article shows the process of cross-cultural adaptation of the RKPPS for the Spanish-speaking population of an observational play assessment. As in other European countries, there is a scarce number of assessments that assess play skills, which reduces their assessment to unstructured observation, the use of developmental inventories, or the use of tests with normative data from other cultural contexts [10]. Given that cultural adaptation is essential when an assessment test is used in a different context than that of the original version, we considered that the resulting RKPPS translation has conceptual and semantic equivalence to the original English version, which reduces the bias of the study [21]. Moreover, this process is necessary to ensure that the adapted test is a valid version of the original test and therefore can be used in the target population. Currently, there are no data available to report on the main assessment that Spanish occupational therapy professionals use to observationally evaluate children’s play, so there are no standards or cultural adaptations available for Spanish-speaking countries. Therefore, to avoid harming the new cultural group in which the test is to be used, it is advisable to administer a culturally adapted test to the new context [27]. In addition, due to the scarce availability of assessments based on the observation of playing skills, the cultural adaptation of the RKPPS makes it easier for Spanish-speaking countries to access a test with consistent psychometric properties [28].

Adapting an existing test, rather than developing a new one applicable to the target population, has several advantages for the researcher as it allows comparison of data from different population groups and different contexts. Culturally adapted tests allow for the generalization of results and identifying and addressing contextual differences between study populations. In the current literature, there are only two studies available that have carried out a process similar to the one developed in the present research. In this case, the studies conducted by Pacciuilio et al. [29] and Sposito et al. [30] provided data on the process of cultural adaptation and inter-rater reliability of the Portuguese translated version of the RKPPS. Compared to the study by Pacciulio et al. [29] that included two phases of translation and peer review, in our research an additional comprehensibility analysis process was incorporated into the methodological phases. In this case, the methodology we used followed the recommended guidelines for cultural adaptation proposed by the ITC [14] and Muñiz et al. [20]. Using this methodology provides a valuable model for cultural adaptation of assessment tests by including stakeholders (e.g., occupational therapists) in the adaptation process, thus ensuring a better degree of fit of the translated version. Furthermore, regarding inter-rater reliability analysis, our findings were similar to those obtained by Sposito et al. [30], evidencing an adequate degree of agreement for the items and item categories of the adapted version of the RKPPS.

Thus, this research followed a cultural adaptation methodology similar to that used by other researchers, such as Romero-Ayuso et al. [31] who adapted the My Child’s Play questionnaire into Spanish, a self-completed questionnaire by parents, or the study by Montes-Montes et al. [32] who adapted the Developmental Coordination Disorder Questionnaire (DCDQ–ES) into Spanish. Furthermore, in our research, the incorporation of a panel of experts added greater rigor to the adaptation process, allowing the result of the cultural adaptation to be useful and understandable.

### Limitations and Future Research

This study has several limitations. Firstly, a convenience sample limited the number of participants, so some selection bias may have occurred. Secondly, although this study is preliminary, expanding the sample size and incorporating a sample of children with neurodevelopmental disorders to evaluate the validity and discriminatory capacity of the test in the population with pathology would be appropriate. The translated version is adjusted to the Spanish cultural context, so it is necessary to make the necessary grammatical and semantic changes in case it is used in other Spanish-speaking countries to ensure appropriateness in a different cultural context. As future lines of research, additional psychometric analysis to confirm the validity and usefulness of the adapted test should be performed.

## 5. Conclusions

The RKPPS Spanish version provides observational assessment of children’s play skills; especially, it allows us to identify the level of development and skill shown by the child during a free play situation. The RKPPS allows the assessment of play skills from a child’s performance-centered viewpoint in such a way that it allows us to identify which skills may be affected and may hinder the child from participating adequately in free play.

## Figures and Tables

**Figure 1 children-10-00965-f001:**
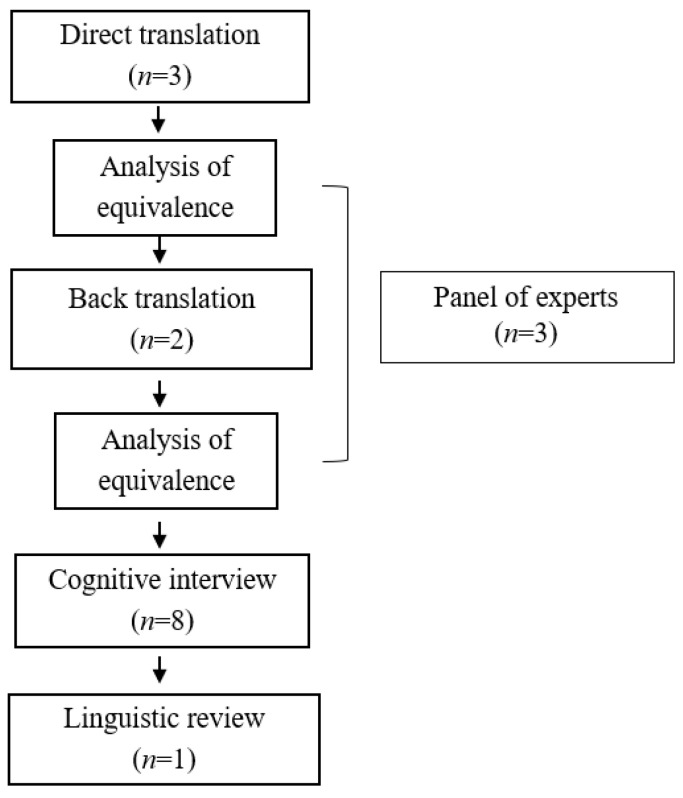
Cultural adaptation procedure.

**Table 1 children-10-00965-t001:** Examples of direct/back translation procedures.

Original Test Item	Spanish Translation	Modifications	Final Translation
Beginning integration of entire body in activities. Concentrates on complex movements, jumps off floor, stands on one foot briefly, throws ball in stance without falling.	Comienza a interactuar con todo el cuerpo en actividades, se concentra en movimientos complejos, salta, se para en un pie brevemente, lanza una pelota sin caerse.	Replace “se pone en un pie” to “se pone a la pata coja”	Comienza a interactuar con todo el cuerpo en actividades, se concentra en movimientos complejos, salta, se pone a la pata coja, lanza una pelota sin caerse.
Runs around obstacles, turns corners, climbs nursery apparatus, walks up and down stairs (alternating feet), catches ball by trapping it, stands on tiptoe.	Corre alrededor de los obstáculos, gira en las esquinas; trepa estructuras de juego; sube y baja escaleras (alternando pies); agarra y atrapa las pelotas; se pone sobre las puntas de los pies.	Replace “se pone en las puntas de los pies” to “se pone de puntillas”	Corre alrededor de obstáculos, gira en las esquinas; trepa estructuras de juego; sube y baja escaleras (alternando pies); agarra y atrapa las pelotas; se pone de puntillas.
Quiet play 5 to 10 min; play with single object 5 min.	Juego tranquilo por 5 a 10 min, juego con un objeto por 5 min.	Replace “por” to “durante”. Add “se observa” at the beginning of the item	Se observa juego tranquilo durante 5 a 10 min; juego con un objeto durante 5 min.
Imitation of observed facial expressions and physical movement (i.e., smiling, pat-a-cake), imitates vocalizations.	Imitación de expresiones faciales que observa y de movimientos (p. ej., sonrisas, acariciar un pastel), imita vocalizaciones.	Replace “acariciar un pastel” to “palmas palmitas”	Expresiones faciales y de movimientos que observa (p. ej., sonrisas, palmas palmitas); imita vocalizaciones.

**Table 2 children-10-00965-t002:** Examples of cognitive interviews process.

Original Test Item	Translation	Rationale	Final Translation
Runs, squats, climbs on and off chairs, walks up and down stairs (step to gait), kicks ball, rides kiddy car.	Corre; se pone en cuclillas; trepa y se baja de las sillas; sube y baja escaleras sin alternancia; patea pelotas; conduce un coche con las piernas.	There were no comprehension difficulties. Even so, five participants suggest changing “patea pelotas” to “da patadas a una pelota”	Corre; se pone en cuclillas; trepa y se baja de las sillas; sube y baja escaleras sin alternancia; da patadas a una pelota; conduce un coche con las piernas.
15 s for detailed object, 30 s for visual and auditory toy.	15 segundos para los objetos con detalle; 30 segundos para juguetes visual o auditivo.	Three participants suggest completing the items by including “se mantiene atento” to improve its comprehensibility	Se mantiene atento 15 segundos para los objetos con detalle; 30 segundos para juguetes visual o auditivo.
Attends to sounds and voices, babbles, uses razzing sounds.	Presta atención a los sonidos y voces, balbucea, hace sonidos con la lengua entre labios.	Four participants suggest changing the expresión “hace sonido con la lengua entre los labios”, to “hace pedorretas”	Presta atención a los sonidos y voces; balbucea; hace pedorretas.
Combination of solitary and onlooker, beginning interaction with peers.	Combinación de juego solitario y espectador, comienza la interacción con niños compañeros.	Six participants suggest changing the term “niños compañeros” to “iguales”	Combinación de juego solitario y espectador; comienza la interacción con sus iguales.
More complex games with a variety of adults (hide and seek, chasing), commands others to carry out actions.	Juegos más complejos con distintos adultos (a esconderse; persecuciones); ordena a otros que realicen acciones.	Three participants suggest the need to modify the expressions “a esconderse” to “jugar al escondite”, and “persecuciones” to “jugar al pillapilla”	Juegos más complejos con distintos adultos (p. ej., jugar al escondite; jugar al pillapilla); ordena a otros que realicen acciones.

**Table 3 children-10-00965-t003:** Inter-rater reliability analysis.

Space Management		Material Management		Pretense/ Symbolism		Participation	
Item	κ	Item	κ	Item	κ	Item	κ
Gross motor	0.96 ***	Manipulation	0.86 ***	Imitation	0.82 ***	Type	0.89 ***
		Construction	0.93 ***			Cooperation	0.94 ***
Interest	0.94 ***	Purpose	0.92 ***	Dramatization	0.87 ***	Humor	0.87 ***
		Attention	0.91 ***			Language	0.82 ***

Note: κ: Cohen’s Kappa index; *** *p* < 0.001.

**Table 4 children-10-00965-t004:** Cohen’s Kappa index of RKPPS dimensions and total score.

Item	κ
Space Management	0.93 ***
Material Management	0.91 ***
Pretense/Symbolism	0.81 ***
Participation	0.89 ***
Total Score	0.90 ***

Note: κ: Cohen’s Kappa index; *** *p* < 0.001.

## Data Availability

The data presented in this study are available on request from the corresponding author. The data are not publicly available due to privacy reasons.

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
