# Peer review of "Spanish Cultural Adaptation and Inter-Rater Reliability of the Revised Knox Preschool Play Scale"

_children, 2023, doi:10.3390/children10060965_

Round 1

Reviewer 1 Report

The article presents a study  that examines the validity of an assessment tool for use in Spanish speaking participants. The authors did a very good job of the research and I am only suggesting minor changes. 

The article would be improved if the authors made a stronger argument on the value of conducting this type of work. It is not overly original, but it is necessary if researchers are to be able to trust their tools, and if the tools are not trustworthy, then theory cant be appropriately tested. I would like to see more of this type of research published as all too often validity checks are the awkward job no-one wants to do and then findings don’t go as planned, and then we don’t know if it was due to the measure not being suitable for the population on which it is being used. This revision could occur in the introduction and discussion.

Some specifics..

In the abstract the author introduce the term a structured assessment test? The paper would be easier to follow if this was explained.

It is 3 groups and an individual – one person is not a group. Group is a word to describe more than one person. You have 4 classifications of participants with some classifications having groups and one classification being a single individual.

Reviewer 2 Report

Overall I think the paper is interesting. However, there appear to be many problems with the method. Of concern are the small sample size and the method of statistical analysis. In particular, the statistical analysis seems to need to be reanalyzed. If inter-rater reliability and validity are to be assessed, perhaps Cohen's Kappa should be used for the analysis. Therefore, we conclude that significant revisions are necessary.

Round 2

Reviewer 2 Report

The authors made necessary revisions.

The manuscript is ok for acceptance.

Author Response

Dear Reviewer,

Thank you for your comments to improve the quality of the manuscript. We appreciate that you have considered the paper suitable for publication. 

Sincerely,

The authors